Five new malformed trilobites from Cambrian and Ordovician deposits from the Natural History Museum

Bicknell Russell D.C. rdcbicknell@gmail.com 1 2
Smith Patrick M. 3 4
1 American Museum of Natural History , New York City , NY , United States of America
2 School of Environmental and Rural Science, University of New England , Armidale , New South Wales , Australia
3 Palaeontology Department, Australian Museum Research Institute , Sydney , New South Wales , Australia
4 Department of Biological Sciences, Macquarie University , Sydney , Australia
Lieberman Bruce
Electronic publication date: 2023 Oct 26
Publication date: 2023
Volume: 11
Electronic Location ID: e16326
Received 2023 Aug 24; Accepted 2023 Sep 30
Copyright: ©2023 Bicknell et al.
Copyright year: 2023
Copyright holder: Bicknell et al.
License: This is an open access article distributed under the terms of the Creative Commons Attribution License, which permits unrestricted use, distribution, reproduction and adaptation in any medium and for any purpose provided that it is properly attributed. For attribution, the original author(s), title, publication source (PeerJ) and either DOI or URL of the article must be cited.
License URL: https://creativecommons.org/licenses/by/4.0/

Keywords: Trilobites, Injuries, Predator-prey systems, Predation, Paleozoic, Burgess Shale, Jince formation, Llanfawr Mudstones Formation, Meadowtown Formation

Funding: Australian Research Council DP200102005 University of New England Postdoctoral Fellowship American Museum of Natural History Postdoctoral Fellowship This research was funded by an Australian Research Council grant (DP200102005), a University of New England Postdoctoral Fellowship (to Russell DC Bicknell), and an American Museum of Natural History Postdoctoral Fellowship (to Russell DC Bicknell). The funders had no role in study design, data collection and analysis, decision to publish, or preparation of the manuscript.

==============================
Injured trilobites present insight into how a completely extinct group of arthropods responded to traumatic experiences, such as failed predation and moulting complications. These specimens are therefore important for more thoroughly understanding the Paleozoic predator-prey systems that involved trilobites. To expand the record of injured trilobites, we present new examples of injured Ogygopsis klotzi and Olenoides serratus from the Campsite Cliff Shale Member of the Burgess Shale Formation (Cambrian, Miaolingian, Wuliuan), Paradoxides (Paradoxides) paradoxissimus gracilis from the Jince Formation (Cambrian, Miaolingian, Drumian), Ogygiocarella angustissima from the Llanfawr Mudstones Formation (Middle–Late Ordovician, Darriwilian–Sandbian), and Ogygiocarella debuchii from the Meadowtown Formation, (Middle–Late Ordovician, Darriwilian–Sandbian). We consider the possible origins of these malformations and conclude that most injuries reflect failed predation. Within this framework, possible predators are presented, and we uncover a marked shift in the diversity of animals that targeted trilobites in the Ordovician. We also collate other records of injured Ogygo. klotzi and Ol. serratus, and Ogygi. debuchii, highlighting that these species are targets for further understanding patterns and records of trilobite injuries.

Introduction

Numerous injured trilobites have been reported from Cambrian to Devonian aged deposits (Owen, 1985; Babcock, 1993; Bicknell et al., 2022d). These malformations have presented insight into the position of trilobites as prey items (Rudkin, 1979; Conway Morris & Jenkins, 1985; Owen, 1985; Zong, 2021b; Bicknell et al., 2022d), as well as how trilobites recovered from moulting complications (McNamara & Rudkin, 1984; Owen, 1985). Such specimens are therefore useful for understanding the palaeoecology and palaeobiology of the completely extinct arthropod group (Owen, 1985; Rudkin, 1985; Babcock, 1993).

Trilobite injuries are considered exoskeletal breakage from accidental injury, attack, or moulting issues (Bicknell et al., 2022a). Injuries are generally ‘L’-, ‘U’-, ‘V’-, or ‘W’-shaped indentations (Babcock, 1993; Bicknell & Pates, 2019; Bicknell et al., 2022a) and can also be expressed as rounded and reduced exoskeletal sections, or as a ‘single segment injury’ (SSI; Pates et al., 2017; Pates & Bicknell, 2019; Bicknell et al., 2022a; Bicknell et al., 2022e). Injuries commonly show evidence for cicatrisation and/or segment repair and regeneration—records of a successful moulting after an injury. Occasionally, abnormal structures, such as fusion of exoskeletal sections or lack of segment expression, are associated with injuries (Owen, 1985; Bicknell et al., 2022a; Bicknell et al., 2023a). This reflects abnormal recovery from the injury. Importantly, these morphologies differ from teratologies that record how trilobites responded to genetic or developmental malfunctions (Owen, 1985; Babcock, 2007; Bicknell & Smith, 2021; Bicknell & Smith, 2022).

To expand the record of injured trilobites from lower Paleozoic deposits, five novel specimens from the Natural History Museum Invertebrate palaeontology collection (NHMUK PI) that have injuries are reported here. These are specimens of Ogygiocarella angustissima (Salter, 1865), Ogygiocarella debuchii (Brongniart, 1822), Ogygopsis klotzi (Rominger, 1887), Olenoides serratus (Rominger, 1887), and Paradoxides (Paradoxides) paradoxissimus gracilis (Boeck, 1827). We also collate other evidence of malformed specimens of these species to explore possible injury causes and patterns.

Geological context

The Ogygopsis klotzi (NHMUK PI I 4749) and Olenoides serratus (NHMUK PI IG 4437-9) figured in this study were collected from Mount Stephen in British Columbia, Canada, in Walcott’s (1908) “Ogygopsis Shale”, Burgess Shale Formation on the mountain trail 850 m above the town of Field. This is horizon is now placed within the Campsite Cliff Shale Member, a member that also outcrops at Mount Field and the Fossil Gully Fault (Fletcher & Collins, 1998; Fletcher & Collins, 2003). The association of articulated trilobite remains and with an underlying distal wedge facies suggests the unit was deposited in a deeper water, potentially euoxic setting (further from the carbonate platform forming the Cathedral Formation palaeocliff edge; Allison & Brett, 1995). Presence of the eponym for the Pagetia bootes Subzone places the member firmly within the restricted shelf Bathyuriscus–Elrathina Zone (Fletcher & Collins, 1998). This has been correlated with the upper portion of the open-shelf Ptychagnostus praecurrens Zone of North America (Robison & Babcock, 2011) and is correlated with the later portion of the Wuliuan Stage (Miaolingian) on the global scale (Peng, Babcock & Cooper, 2012).

The Paradoxides (Paradoxides) paradoxissimus gracilis (NHMUK PI OR 42440) figured here was collected from Jince in the Czech Republic at an unknown locality in the Jince Formation. The taxon is widely distributed over the entire Jince Basin, occurring at multiple outcrops throughout the Litavka River Valley region, hence the exact specimen location is impossible to determine (Fatka & Szabad, 2014). However, more generally P. (P.) paradoxissimus gracilis occurs in the green, fine-grained greywackes and shales within the middle levels of the Jince Formation, which is thought to correspond with the peak of a transgressive event identified in the unit (Storch, Fatka & Kraft, 1993; Fatka & Szabad, 2014). The occurrence of abundant articulated agnostids, paradoxidids, and a conocoryphid species (Fatka, Kordule & Szabad, 2004) suggests a deeper water environment, within the conocoryphid biofacies of Álvaro, González-Gómez & Vizcaïno (2003). The taxon is the eponym for the P. (P.) paradoxissimus gracilis Zone in the Příbram-Jince Basin within the Barrandian area (Fatka & Szabad, 2014 for a comprehensive discussion). Co-occurrence of the agnostid Hypagnostus parvifrons (Linnarsson, 1869) and the lower stratigraphic occurrence of Onymagnostus hybridus (Brøgger, 1878; Fatka, Kordule & Szabad, 2004; Fatka & Szabad, 2014) suggests the zone corresponds with the middle and higher levels of the Baltic P. (P.) paradoxissimus Zone (Axheimer & Ahlberg, 2003; Høyberget & Bruton, 2008; Weidner & Nielsen, 2014). This region likely corresponds to the Scandinavian, South Chinese, and Australian Ptychagnostus atavus and Ptychagnostus punctuosus zones, placing the formation in the early to mid-Drumian Stage (Miaolingian) globally (Peng, Babcock & Cooper, 2012).

The Ogygiocarella angustissima (NHMUK PI OR 59206) and Ogygiocarella debuchii (NHMUK PI In 23066) figured in this study were collected from two similarly aged deposits in Wales, United Kingdom. The Ogygiocarella angustissima specimen was apparently collected from Gwern-fydd, although this seems unlikely given the regional geology and lithology of the specimen. Lectotype material preserved in identical matrix has alternatively been suggested to be derived from “Harper’s quarry”, 500 m north-west of Welfield(near Builth), likely within the Llanfawr Mudstones Formation, Builth Inlier (Hughes, 1979 and references therein, also see updated stratigraphy in Davies et al., 1997; Fortey et al., 2000). The Ogygiocarella debuchii specimen conversely originates from Betton Quarry (near Shropshire) in the upper Meadowtown Formation, Shelve Inlier. Both units are dominated by fine mudstone and siltstone, likely being deposited on a relatively deep shelf environment nearby a volcanic arc (Fortey & Owens, 1987; Davies et al., 1997; Fortey et al., 2000; Owens, 2002). Known ranges of Ogygiocarella angustissima and Ogygiocarella debuchii suggest the taxa occur in either the Hustedograptus teretiusculus and/or Nemagraptus gracilis zones at these localities. However, without more precise details regarding the exact collection horizons (and associated graptolite or other shelly fauna) it impossible to determine which precisely (Hughes, 1979; Bettley, Fortey & Siveter, 2001). Hence, the figured material likely comes from somewhere within the regional Llandeilian(Llanvirn) or Aurelucian (Caradoc) stages. This correlates with the global Darriwilian (Middle Ordovician) to Sandbian (Late Ordovician) boundary (Bettley, Fortey & Siveter, 2001; Cooper & Sadler, 2012; Goldman et al., 2023).

Methods

Trilobite specimens within the Natural History Museum Invertebrate palaeontology collection, London were reviewed for injuries. Identified specimens were from the Campsite Cliff Shale Member of the Burgess Shale Formation, Canada; Jince Formation Czech Republic; and the Llanfawr Mudstones and Meadowtown formations, Wales, UK. These specimens were photographed under low angle LED light as stacks with a Canon EOS 600D at the NHM. Images were stacked using Helicon Focus 7 (Helicon Soft Limited) stacking software. Measurements of specimens were collated from the images using ImageJ (Schneider, Rasband & Eliceiri, 2012) and compiled into Table 1.

Results

Ogygopsis klotzi (Rominger, 1887), NHMUK PI I 4749, Cambrian (Miaolingian, Wuliuan), Campsite Cliff Shale Member of the Burgess Shale Formation, Canada. Figs. 1B and 1E.

Figure 1 Injured Olenoides serratus (Rominger, 1887) and Ogygopsis klotzi (Rominger, 1887) from the Cambrian (Miaolingian, Wuliuan) aged Campsite Cliff Shale Member, Burgess Shale Formation, Canada.

(A, C, D) Olenoides serratus. NHMUK PI IG 4437-9. (A) Complete specimen. (C) Close up anterior injury showing truncated (white arrow) and recovering (black arrow) pleural spines. (D) Close up of ‘U’-shaped injury showing limited cicatrisation. (B, E) Ogygopsis klotzi. NHMUK PI I 4749. (B) Complete specimen. (E) Close up of ‘U’-shaped injury (white arrows). (A, C, D) Images converted to greyscale.

NHMUK PI I 4749 is a partial, moulted, internal exoskeletal mould with an injury on the right thoracic pleural lobe. The injury is an asymmetric ‘U’-shaped indentation that truncates the 5th and 6th pleural spines by a maximum of 6.6 mm. The 5th spine shows limited pleural spine recovery, and the 6th pleural spine is rounded.

Olenoides serratus (Rominger, 1887), NHMUK PI IG 4437-9, Cambrian (Miaolingian, Wuliuan) Campsite Cliff Shale Member of the Burgess Shale Formation, Canada Figs. 1A, 1C and 1D.

Table 1 Measurements of documented specimens.

Species	Specimen number	Stage	Cephalic length (mm)	Cephalic width (mm)	Thoracic length (mm)	Pygidial length (mm)	Pygidial width (mm)	Figure number	
Ogygopsis klotzi	NHMUK PI I 4749	Cambrian (Miaolingian, Wuliuan)	22.1	46.1*	27.3	29.3	37.9*	Figs. 1B and 1E	
Olenoides serratus	NHMUK PI IG 4437-9	Cambrian (Miaolingian, Wuliuan)	21.6*	36.9*	24.2	16.4*	32.7	Figs. 1A, 1C and 1D	
Paradoxides (Paradoxides) paradoxissimus gracilis	NHMUK PI OR 42440	Cambrian (Drumian)	19.1	54.8	42.7*	8.0	6.7	Figs. 2A–2D	
Ogygiocarella angustissima	NHMUK PI OR 59206	Ordovician (Middle–Late Ordovician, Darriwilian–Sandbian)	22.5	65.1	22.3	28.5	53.0*	Figs. 3A and 3B	
Ogygiocarella debuchii	NHMUK PI In 23066	Ordovician (Middle–Late Ordovician, Darriwilian–Sandbian)	–	–	–	11.4	19.5	Figs. 4A and 4B	
Notes.

* Minimal values where the specimen is broken.

– Exoskeletal section is not observed for the specimen.

NHMUK PI IG 4437-9 is a partial, moulted, internal exoskeletal mould with two injuries on the left thoracic pleural lobe. The anterior injury is a ‘V’-shaped indentation that truncates the 1st and 2nd pleural spines by 4.6 mm and 1.6 mm, respectively. The second pleural spine shows development of a new spine (Fig. 1C). The posterior injury is an asymmetric, cicatrised ‘U’-shaped indentation that truncates the 6th and 7th pleural spines by 7.0 mm (Fig. 1D).

Paradoxides (Paradoxides) paradoxissimus gracilis (Boeck, 1827), NHMUK PI OR 42440, Cambrian(Miaolingian, Drumian), Jince Formation, Czech Republic. Figure 2.

Figure 2 Injured Paradoxides (Paradoxides) paradoxissimus gracilis (Boeck, 1827) from the Cambrian (Miaolingian, Drumian) aged Jince Formation, Czech Republic.

(A–D) NHMUK PI OR 42440. (A) Complete specimen. (B) Close up of unique injury showing a truncated pleural spine (white arrow) with a notched posterior region (grey arrow), a ‘W’-shaped indentation (black arrow), and a truncated, rounded pleural spine (blue arrow). (C) Close up of ‘W’-shaped indentation. (D) Close up of SSI (black arrow).

NHMUK PI OR 42440 is a mostly complete, internal exoskeletal mould with three injuries along the thorax. One on the left thoracic pleural lobe and two on the right pleural lobe. The anterior-most injury is an SSI that truncates the 1st pleural spine on the left pleural lobe by 3.3 mm (Fig. 2D). The anterior-most injury on the right pleural lobe extends across the ?8th–?10th pleural spines (? denote uncertainty of the segment number as the specimen is broken anteriorly) (Fig. 2B). This injury has a unique morphology. The ?8th pleural spine is rounded and truncated by 2.4 mm and the posterior section of the ?8th segment is truncated further by 4.6 mm. The ?9th pleural spine is truncated by 6.4 mm. The indentations on the ?8th and ?9th pleural spines form a ‘W’-shape. The ?10th pleural spine is truncated by 5.4 mm and shows rounding. The posterior-most injury on the right pleural lobe is a ‘W’-shaped indentation that truncates the ?13th and ?14th pleural spines by 4.3 mm and 4.7 mm, respectively (Fig. 2C).

Ogygiocarella angustissima (Salter, 1865), NHMUK PI OR 59206, Middle–Late Ordovician (Darriwilian–Sandbian), Llanfawr Mudstones Formation, Wales, UK Figure 3.

Figure 3 Injured Ogygiocarella angustissima (Salter, 1865), from the Ordovician (Middle–Late, Darriwilian–Sandbian) aged Llanfawr Mudstones Formation, Wales.

(A, B) NHMUK PI OR 59206. (A) Complete specimen. (B) Close up of ‘U’-shaped (white arrows) and ‘V’-shaped (black arrows) indentations.

NHMUK PI OR 59206 is a mostly complete external mould that has two injuries on the right side (=left pleural lobe in life). The anterior-most injury is a ‘U’-shaped indentation that truncates the 1st–3rd pleural spines by 4.7 mm (Fig. 3B, white arrows). All truncated spines show rounding, and the 3rd pleural spine shows recovery (Fig. 3B, white arrows). The posterior injury is a ‘V’-shaped indentation that truncates the 5th and 6th pleural spines by 2.0 mm and 4.2 mm, respectively. Both pleural spines show rounding (Fig. 3B, black arrows).

Ogygiocarella debuchii (Brongniart, 1822), NHMUK PI In 23066, Middle–Late Ordovician (Darriwilian–Sandbian), upper Meadowtown Formation, Wales, UK. Figure 4.

Figure 4 Ogygiocarella debuchii (Brongniart, 1822) from the Ordovician (Middle–Late, Darriwilian–Sandbian) aged upper Meadowtown Formation, Wales.

(A–C) NHMUK PI In 23066. (A) Complete specimen. (B) Close up of right side showing shallow ‘U’-shaped indentation (white arrow) and the fusion pygidial ribs (black arrow). (C) Close up of left side showing disruption and possible fusion (white arrows) of pygidial ribs and irregular rib sizes.

NHMUK PI In 23066 is an isolated, internal mould of a pygidium with two injuries. The injury on the left side has disrupted ?6th–?9th pygidial ribs (as above, ? denotes uncertainty of rib numbers as the specimen appears broken anteriorly) (Fig. 4C). Ribs are disrupted at the ?6th rib showing evidence of possible fusion with the ?7th rib (Fig. 4C). Additionally, at least the ?8th and ?9th ribs show evidence for irregular borders and inconsistent widths. The injury on the right side is a shallow ‘U’-shaped indentation that extends 0.7 mm from the pygidial border (Fig. 4B). The ?7th and ?8th pygidial ribs on the right side are also fused 0.9 mm from the axial lobe, proximal to the indentation (Fig. 4B).

Discussion

Comparing the injuries documented to previously recorded examples of injured trilobites allows us to propose possible origins for the malformations. The ‘U’- ‘V’-, ‘W’-shaped indentations observed here (Figs. 1A–1E; 2A, 2C; 3A, 3B; 4A and 4B) are comparable to examples of injured Cambrian (see Rudkin, 1979; Babcock, 1993; Babcock, 2007; Pates et al., 2017; Bicknell & Pates, 2020; Zong, 2021a; Zong, 2021b; Bicknell et al., 2022e) and Ordovician (see Ludvigsen, 1977; Babcock, 2007; Zong, 2021a; Bicknell et al., 2022d; Bicknell et al., 2022e) trilobites. These examples are commonly attributed to failed predation. Considering this framework, we propose that the injuries in Ogygopsis klotzi (Figs. 1B and 1E), Olenoides serratus (Figs. 1A, 1C and 1D), and Ogygiocarella angustissima (Fig. 3) reflect unsuccessful predation attempts. This aligns with other records noted within the literature (Rudkin, 1979; Bicknell & Paterson, 2018; Bicknell et al., 2022d). However, explanations for the other specimens are needed.

On Ogygopsis klotzi

Records of injured Ogygopsis klotzi represent valuable insight into possible predator–prey dynamics within the Burgess Shale biota (Table 2). These injuries were originally thought to reflect predation by Anomalocaris canadensis (Whiteaves, 1892) (see discussion in Rudkin, 1979). However, recent three-dimensional (3D) kinematic, biomechanical, and computational fluid dynamic modelling have demonstrated that A. canadensis appendages were ineffective at handling biomineralised prey (De Vivo, Lautenschlager & Vinther, 2021; Bicknell et al., 2023b). More plausible predators are the co-occurring trilobites and other artiopodans that have reinforced gnathobasic spines on walking legs (Whittington, 1980; Bruton, 1981; Bicknell et al., 2018b; Holmes, Paterson & García-Bellido, 2020). Trilobite fragments within the gut contents of artiopodans (Zacaï, Vannier & Lerosey-Aubril, 2016; Bicknell & Paterson, 2018) and 3D biomechanical analyses of gnathobase-bearing appendages (Bicknell et al., 2018a; Bicknell et al., 2021) support this mode of durophagous predation. One other option is the mantis shrimp-like arthropod Yohoia Walcott, 1912 that may have damaged trilobite exoskeletons using its anteriorly directed raptorial appendages (Pratt, 1998; Haug et al., 2012).

The records collated in Table 2 represent the basis for developing a much larger dataset to explore Ogygopsis klotzi injury patterns. Documentation of injured specimens housed in other collections will expand this preliminary sample and permit the left–right behavioural asymmetry hypothesis to be re-addressed (Babcock & Robison, 1989; Babcock, 1993). Recent examination of injury patterns in Cambrian trilobites have demonstrated little evidence for injury asymmetry (Pates & Bicknell, 2019; Bicknell et al., 2022a; Bicknell et al., 2023a). However, with 80% of Ogygo. klotzi unilateral injuries being right sided, this injury distribution may indeed reflect a population-level pattern (Table 2). Illustrating this condition with a statistical dataset of one species (following Pates et al., 2017; Bicknell, Paterson & Hopkins, 2019; Bicknell et al., 2022a; Bicknell et al., 2023a; Pates & Bicknell, 2019) will uncover interesting injury patterns and represents a clear direction for exploring this topic further.

Table 2 Summary of injured and malformed Ogygopsis klotzi documented within the literature.

Citation	Injury location	Injury side	Injury morphology	
Rudkin (1979, fig. 1A, B)	Thorax, segments 3–6	Right	‘U’-shaped	
Rudkin (1979, fig. 1C, D)	Thorax, segments 6–8	Left	‘W’-shaped	
Rudkin (1979, fig. 1E, F)	Anterior pygidium	Right	‘W’-shaped	
Rudkin (1979, fig. 1G and 1H), refigured in Rudkin (2009, fig. 1B)	Thorax, segments 7–8	Right	‘V’-shaped	
Briggs & Whittington (1985, p. 37)	Thorax, segment 10, extends into anterior pygidium	Right	‘U’-shaped	
Pratt (1998, fig. 1A)	Anterior pygidium	Right	‘W’-shaped	
Nedin (1999, fig. 2C)	Thorax, segments 3–6	Bilateral	Left: ‘U’-shaped (segment 5). Right: ‘W’-shaped (segments 3–6)	
Bicknell & Pates (2020, fig. 7A, B)	Thorax, segments 2–5	Left	‘W’-shaped	
Bicknell & Holland (2020, fig. 2A, C)	Thorax, segments 1–4	Right	‘L’-shaped	
Bicknell & Holland (2020, fig. 2B, D)	Thorax, segments 5–7	Right	‘U’-shaped with pinched and warped segments	
This article, Figs. 1B and 1D	Thorax, segments 5–6	Right	‘U’-shaped	

On Olenoides serratus

Predators of Olenoides serratus were similar to Ogygopsis klotzi as both species are from the Burgess Shale. It is worth considering how the Ol. serratus male mating claspers may have caused injuries (Losso & Ortega-Hernández, 2022). In modern female horseshoe crabs, males cause injuries to the medial region of during amplexus (Shuster Jr, 1982; Brockmann, 1990; Shuster Jr, 2009; Bicknell, Pates & Botton, 2018c; Das et al., 2021; Bicknell et al., 2022c). It is possible that male Ol. serratus may have caused similar medial injuries during mating. However, these reduced appendages did not produce the large, laterally located injuries documented here and in Table 3.

On Paradoxides (Paradoxides) paradoxissimus gracilis

The anterior indentations on the right pleural lobe of Paradoxides (Paradoxides) paradoxissimus gracilis (NHMUK PI OR 42440, Fig. 2B) reflect either two separate attacks that targeted the same exoskeletal region, or an additional moulting complication proximal to this injury. We suggest that both options are viable here as the posterior-most section of the injury (Fig. 2B, blue arrow) shows more recovery than anterior region. As trilobites recovered from injuries anterior to posterior (McNamara & Tuura, 2011; Zong & Bicknell, 2022), the more anterior region should show more evidence of recovery. This injured region has therefore experienced an additional traumatic event, although the exact cause is unknown. The posterior injury on the right pleural lobe (Fig. 2D) is morphologically comparable to injuries ascribed to predation (Babcock, 1993; Bicknell & Paterson, 2018) supporting the assignment of this injury to failed predation. The SSI observed on this specimen (Fig. 2D) reflects failed predation (Bicknell et al., 2022a) or a moulting complication. As P. (P.) paradoxissimus gracilis has long pleural spines, that may have been damaged while moulting, resulting in an isolated injury (Šnajdr, 1978; Conway Morris & Jenkins, 1985; Daley & Drage, 2016; Drage, 2019; Drage & Daley, 2016). This specimen highlights that more research into the moulting patterns of P. (P.) paradoxissimus gracilis may help differentiate these options.

Table 3 Summary of injured Olenoides serratus documented within the literature.

Citation	Injury location	Injury side	Injury morphology	
Pratt (1998, fig. 1B), refigured in Butterfield (2001, Fig. 1.2.3.1.i)	Genal spine; thorax, segments 1–3	Left	‘W’-shaped	
Bicknell & Paterson (2018, fig. 1F)	Thorax, segments 5–7	Left	‘V’-shaped	
This article, Figs. 1A, 1C and 1D	Thorax, segments 1–2, 6–7	Left	‘V’-shaped (segments 1–2); ‘U’-shaped (segments 6–7)	

Previously documented injured specimens of “P. gracilis” (Boeck, 1827) permit useful comparisons to understand the injuries observed here (Fig. 2; Šnajdr, 1978; Owen, 1985; De Baets et al., 2022). Injuries to Jince Formation Paradoxides are considered a result of moulting complications that arose from the flat morphology and elongate pleural spines common to Paradoxides (Šnajdr, 1978). However, as demonstrated here, failed predation cannot be fully discounted. The predators were likely co-occurring paradoxidids (Babcock, 1993; Fortey & Owens, 1999; Fatka, Szabad & Budil, 2009)—a proposal that further supports cannibalism within Cambrian trilobites (Conway Morris & Jenkins, 1985; Daley et al., 2013; Bicknell et al., 2022a). Additionally, the bivalved arthropod Tuzoia Walcott, 1912 may have targeted Paradoxides, as Tuzoia is considered a nektobenthic to pelagic predatory or scavenger (Fatka & Herynk, 2016; Izquierdo-López & Caron, 2022).

On Ogygiocarella

The abnormal recovery and fusion of ribs in Ogygiocarella debuchii (NHMUK PI In 23066; Fig. 4) in two pygidial regions indicates two different traumatic events. The injury on the left side shows no evidence of an indentation (Fig. 4C). This suggests that a moulting complication occurred, and the ribs recovered abnormally—a condition that was propagated through subsequent moulting events. Conversely, the ‘U’-shaped indentation and fused pygidial ribs on the right side (Fig. 4B) indicates a failed predation attempt that recovered abnormally.

Records of injured and malformed Ogygiocarella are limited(Table 4). However, the identification of five injured specimens since 2022 (Table 4) demonstrates that Ogygiocarella, specifically Ogygi. debuchii, represents another avenue for future research into injury patterns. There is also mounting evidence to support at least three distinct arthropods groups that could have targeted Ogygiocarella as prey. (1) The Middle Ordovician (Darriwilian) Castle Bank Biota fauna (Botting et al., 2023) includes a yohoiid-like arthropod that could have attacked these trilobites using raptorial appendages (Botting et al., 2023). (2) Ordovician eurypterids—forms known from Late Ordovician(Sandbian) aged Welsh deposits (Størmer, 1951; Tetlie, 2007)—have been highlighted as possible, albeit ineffective, predators of trilobites (Lamsdell et al., 2015; Bicknell, Melzer & Schmidt, 2022b; Schmidt et al., 2022). If they were the predators, eurypterids would have targeted trilobites during a soft-shelled stage. (3) The large asaphid trilobites themselves could have targeted each other and used gnathobasic spines on walking legs to process the biomineralised exoskeletons.

Table 4 Summary of injured Ogygiocarella within the literature.

Citation	Species	Injury location	Injury side	Injury morphology	
Bicknell et al. (2022d, fig. 4a, b)	Ogygiocarella debuchii	Cephalon and genal spine	Right	Truncated genal spine, ‘U’-shaped along posterior margin of spine	
Bicknell et al. (2022d, fig. 4c, d)	Ogygiocarella debuchii	Pygidium	Left	‘U’- and ‘V’-shaped	
Bicknell et al. (2022e, fig. 2.1, 2.2)	Ogygiocarella debuchii	Pygidium	Left	‘W’-shaped	
This article, Figs. 3A and 3B	Ogygiocarella angustissima	Thorax, segments 1–3, 5–6	Left (right in the counterpart)	‘U’-shaped (segments 1–3), ‘V’-shaped (segments 5–6)	
This article, Figs. 4A–4D	Ogygiocarella debuchii	Pygidium	Bilateral	Left: Disrupted and fused ribs. Right: ‘U’-shaped and fused ribs	

Beyond arthropods, nautiloids are commonly suggested as Ordovician predators of trilobites (Brett, 2003; Klug et al., 2018). These large cephalopods would have been able to grapple trilobites with tentacles and damage the exoskeletons with re-enforced beaks (Klug et al., 2018).

We thank Richard Howard for access to, and assistance with the NHM collection. We also thank Richard Fortey for his assistance in identifying NHMUK PI In 23066. Finally, we thank Danita Brandt and Oldřich Fatka for their constructive reviews that improved the manuscript.

Additional Information and Declarations

Competing Interests

Author Contributions

Data Availability

The authors declare there are no competing interests.

Russell D.C. Bicknell conceived and designed the experiments, performed the experiments, analyzed the data, prepared figures and/or tables, authored or reviewed drafts of the article, and approved the final draft.

Patrick M. Smith conceived and designed the experiments, analyzed the data, authored or reviewed drafts of the article, and approved the final draft.

The following information was supplied regarding data availability:

The specimens are all housed within the Natural History Museum, London:

- NHMUK PI I 4749

- NHMUK PI IG 4437-9

- NHMUK PI OR 42440

- NHMUK PI OR 59206

- NHMUK PI In 23066

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
