# Peer review of "Five new malformed trilobites from Cambrian and Ordovician deposits from the Natural History Museum"

_PeerJ, doi:10.7717/peerj.16326_

## Round 0.1 · original submission · Minor Revisions

The reviews for your paper are now in and they have recommended minor revisions are necessary before publication and I concur with their assessments. The reviewers have provided a number of very useful comments and suggestions and when preparing your revision, please pay careful attention and be sure to address the points they describe. Note, both reviewers have suggested a change to the title and I would incorporate their suggestions on that as well. When you submit your revision, include a version showing the changes made tracked, along with a detailed letter explaining the changes you have made. If there are any changes you have not made please detail those as well.

·

Basic reporting

It would be helpful to explain the museum numbers/abbreviations when they are first used in the ms rather than waiting until the "Methods" section.

Related comment--it's not clear until the methods section that the specimens were from museum collections. The "Geological Context" section confuses things by seeming to describe field collections.

Editorial comments
Text line 161: Awkward wording, replace "Additionally, more posterior ribs have irregular borders..." with specific number of affected ribs

Text line 179, the article "the" is missing from "As trilobite recovered"

Experimental design

The 'experimental design' is comparing the trilobite malformations to previously described examples and inferring whether the malformations are due to predation or moulting. There are no new insights (e.g., criteria or new tools/methods) offered in making this distinction.

Validity of the findings

There is value in adding these 5 specimens to the catalog of trilobites showing malformations. However, the authors cannot state definitively whether the malformations are from predation or moulting (lines 173-174, 182-186207-208,213-215, 228-229). There are many "could haves" in the text. We don't really learn anything new, and the paper is unsatisfying in that regard. Reference to the GOBE comes out of left field as a 'big-picture' tie-in and, without a more extensive discussion, is not supportable, given the uncertainty of the origins of trilobite malformations and small sample size of malformed trilobites..

Additional comments

I advocate changing the word "injured", in the title, which carries the connotation of harm (perhaps these malformations were benign!) to "malformations" in the sense of Babcock, 1993. The main contribution of that paper is adding specimens to the list of known trilobites with malformations. As such, this could be a much shorter note comprising (1) the specimen descriptions, (2) photos and (3) data tables. The information in the "Geological context" section does not contribute to an analysis that might give insight into the origin of the malformations nor is this information used in a comparative sense, and, as mentioned above, the concluding sentence (lines 250-252) is an overreach. .The paper can easily be re-envisioned by "minor" editing--focusing on the three elements listed above.

·

Basic reporting

The contribution is written in a clear scientific style conforming to professional standards. The text is adequately introduced by a brief review of the discussed problematics, and relevant geologic and stratigraphic documentation for all studied samples. The discussed material is properly included within broader context and is supported by adequate properly cited references to earlier published studies. All figures are relevant to the content and are of a good quality and resolution, all are appropriately labelled and referred in text.
The explanation of how described injuries originate are relevant.

In my opinion, it would be adequate to change the title of the contribution into
“Five injured Cambrian and Ordovician trilobites from the Natural History Museum in London”

Experimental design

The study fits well with the scope and aims of PeerJ. The contained data are new and the provided interpretation supplements earlier knowledge about wounded Cambrian and Ordovician trilobites.
The technical standards of methods relevant for such type of investigation are filled; earlier interpretations of comparable Paleozoic materials are properly discussed and referred.

Validity of the findings

This contribution provides new information on anomalous trilobites; morphological changes of trilobite exoskeletons are properly documented and discussed. Reported specimens are housed in the Natural History Museum Invertebrate palaeontology collection (NHMUK), London. There is no speculation, all conclusions are well stated and linked to studied specimens which support all results.

Additional comments

The paper represents an interesting and important contribution bringing new data (and adequate interpretation) and is worth to be printed in PeerJ.
At several places I did remarks asking some more details; these are to be discussed in letter to editor.

---

## Round 0.2 · accepted · Accept

The authors have done a good job addressing the comments from the reviewers and in my opinion, the paper is now ready to be accepted.